# Genetic Factors Associated with COPD Depend on the Ancestral Caucasian/Amerindian Component in the Mexican Population

**DOI:** 10.3390/diagnostics11040599

**Published:** 2021-03-27

**Authors:** Gloria Pérez-Rubio, Ramcés Falfán-Valencia, Juan Carlos Fernández-López, Alejandra Ramírez-Venegas, Rafael de Jesús Hernández-Zenteno, Fernando Flores-Trujillo, Irma Silva-Zolezzi

**Affiliations:** 1HLA Laboratory, Instituto Nacional de Enfermedades Respiratorias Ismael Cosío Villegas, Mexico City 14080, Mexico; rfalfanv@iner.gob.mx; 2Computational Genomics Department, Instituto Nacional de Medicina Genómica, Mexico City 14610, Mexico; jfernandez@inmegen.gob.mx; 3Tobacco Smoking and COPD Research Department, Instituto Nacional de Enfermedades Respiratorias Ismael Cosío Villegas, Mexico City 14080, Mexico; aleravas@hotmail.com (A.R.-V.); rafherzen@yahoo.com.mx (R.d.J.H.-Z.); flotfer4011917@hotmail.com (F.F.-T.); 4Instituto Nacional de Medicina Genómica, Mexico City 14610, Mexico; irma.silvazolezzi@rdsg.nestle.com

**Keywords:** COPD, genetic susceptibility, *CHRNA5*, *CHRNA3*, *CYP2A6*, Mexican mestizo, Hispanic paradox

## Abstract

Genetic variability influences the susceptibility to and severity of complex diseases; there is a lower risk of COPD in Hispanics than in non-Hispanic Caucasians. In this study, we included 830 Mexican-Mestizo subjects; 299 were patients with COPD secondary to tobacco smoking, and 531 were smokers without COPD. We employed a customized genotyping array of single nucleotide polymorphisms (SNPs). The population structure was evaluated by principal component analysis and allele association through a logistic regression model and haplotype identification. In this study, 118 individuals were identified with a high Caucasian component and 712 with a high Amerindian component. Independent of the ancestral contribution, two SNPs were associated with a reduced risk (*p* ≤ 0.01) of developing COPD in the *CYP2A6* (rs4105144) and *CYP2B6* (rs10426235) genes; however, a haplotype was associated with an increased risk of COPD (*p* = 0.007, OR = 2.47) in the *CHRNA5-CHRNA3* loci among smokers with a high Caucasian component. In Mexican-Mestizo smokers, there are SNPs in genes that encode proteins responsible for the metabolism of nicotine associated with a lower risk of COPD; individuals with a high Caucasian component harboring a haplotype in the *CHRNA5-CHRNA3* loci have a higher risk of suffering from COPD.

## 1. Introduction

Chronic obstructive pulmonary disease (COPD) is a common, preventable, and treatable condition characterized by dyspnea, chronic cough, and airflow limitation produced by airway or alveolar abnormalities, usually triggered by prolonged exposure to harmful particles or gases. The most common but not the only risk factor is tobacco smoking [1]. This disease results from the complex interaction among chronic tobacco smoking, airway hypersensitivity, inadequate lung development during childhood or neonatal stage, genetic factors, and epigenetic mechanisms, among others [2,3]. However, the prevalence, morbidity, and mortality rates differ across populations [4]; this indicates that the ancestral contribution influences COPD incidence.

An example is the PLATINO (Latin American Pulmonary Obstruction Research Project), which was carried out in five cities in Latin America (Santiago, Chile; Mexico City, Mexico; Montevideo, Uruguay; Caracas, Venezuela; and San Pablo, Brazil). In this study, the prevalence of COPD (adjusting for age, sex, ethnicity, education, pack-years smoked, body mass index, and altitude) was 11.9% in Mexico City and 19.4% in Montevideo [5]. However, these data cannot be generalized for Latin American admixed populations, such as in Mexico, due to the history of the conquest by Spaniards and the introduction of other population groups. According to the geographical regions, there are subpopulations with various ancestral genetic components; in northern Mexican states, individuals with a high Caucasian background predominate, but in southern Mexico, the Amerindian genetic component is higher [6]. We aimed to identify single nucleotide polymorphisms (SNPs) associated with COPD secondary to tobacco smoking in Mexican Mestizos population with a high Amerindian component and compare them with Mexican Mestizos with a high Caucasian component.

## 2. Materials and Methods

### 2.1. Study Population

An observational analytical study was performed; the study population included eight hundred and thirty participants classified into two groups: smokers with COPD (*n* = 299, CTS) and without COPD (*n* = 531, SWOC). All individuals were recruited from the Department of Tobacco Research and COPD (DITABE) of the Mexican Instituto Nacional de Enfermedades Respiratorias Ismael Cosío Villegas (INER) of Mexico City. The inclusion criteria were being Mexican Mestizos by ancestry (parents and grandparents born in Mexico), being over 30 years old, current or former smokers, and smoking ≥10 cigarettes per day for at least ten years. According to the ATS criteria, the diagnosis was based on clinical history, physical examination by a specialized pulmonologist, and spirometry data [7]. The reference values for the Mexican population were used [8]. The postbronchodilator test was performed by administering 400 mg of nebulized salbutamol using an inhaler and spacer [9]. Smokers without COPD (SWOC) came from the smoking cessation support clinic of the same department or from those who attended the “COPD early detection campaigns” in the framework of the celebration of World COPD Day and World No Tobacco Day. Individuals with bronchial asthma, bronchiectasis, active tuberculosis, lung cancer, cystic fibrosis, hypersensitivity pneumonitis, or idiopathic pulmonary fibrosis were excluded from the study. The participants completed a questionnaire regarding demographical data and family history. A 7 mL sample of peripheral blood was collected in EDTA tubes as an anticoagulant. The individuals agreed to participate voluntarily and signed an informed consent document specifically for this protocol, which was previously approved by the INER research and biosafety bioethics committees (protocol code numbers B20-08 and B14-17).

### 2.2. Genomic DNA Extraction and Concentration Adjustment

Genomic DNA was extracted with a commercial BDtract isolation kit (Maxim Biotech, San Francisco, CA, USA). It was quantified by UV light microscopy at 260 nm using a NanoDrop 2000 (Thermo Fisher Scientific, Waltham, MA, USA). A sample of optimum purity was considered when the 260/280 ratio had a value between 1.8–2.0. All samples that met the above criteria were adjusted to 50 ng/µL for subsequent genotyping.

### 2.3. SNPs Selection and Microarray Design

We employed a custom genotyping array, GoldenGate modality (Illumina, Inc., San Diego, CA, USA); we analyzed 452 SNPs in candidate genes, and 253 were ancestry-informative markers (AIMs). This array was used previously in our workgroup to identify genetic variants associated with COPD risk and severity in the Mexican Mestizo population [10]. SNPs selection in candidate genes followed a search strategy for tag SNPs; for this, the Haploview program was used [11]. We established the following criteria: linkage disequilibrium (LD) ≥ 0.8 and minor allele frequency (MAF) ≥ 10% in the Mexican Mestizo population according to the Mexican Genome Diversity Project (MGDP) [6,12] and compliance with the Hardy–Weinberg equilibrium (HWE, *p* ≥ 10^−4^). The candidate genes and number of evaluated SNPs by chromosome are shown in Table 1.

Populational ancestry was evaluated using four reference populations, Caucasian (CEU), East Asian (EA), and African (YRI), of the international HapMap project [13] and Amerindian population of MGDP (NATIVE) [6]; two hundred and fifty-three AIMs were included, which had an allele frequency difference ≥0.4 for each pair of reference populations. Genotypes were obtained through the Illumina GoldenGate platform (Illumina, Inc., San Diego, CA, USA). The microarray was read with a BeadArray scanner of Illumina. Genotypes and documentation generation were performed using GenomeStudio 2011 v. 1.0 software. Samples from individuals who did not meet the call rate criteria (≥95%) were excluded from the analysis.

### 2.4. Statistical Analysis

The SPSS v.20.0 program (SPSS software, IBM, Endicott, NY, USA) was employed to describe the study population. The median, minimum, and maximum values of each continuous numerical variable were obtained, and percentages were reported for the variables sex, and ancestral contribution. Principal component analysis (PCA) was performed using EIGENSOFT v.4.2 software [14], and the population structure was completed with STRUCTURE v.2.3.4 [15] under unsupervised conditions. In both analyses, the populations of HapMap (CEU, YRI, EA) and the NATIVE of the MGDP were treated as a reference. Subsequently, both groups (CTS and SWOC) were separated according to the population structure as a population with a Caucasian predominance (CEU > 50%) or Amerindian predominance (AME > 50%) according to data obtained from the PCA. For each subgroup, allelic association analysis was performed using PLINK 1.07 software [16]; a logistic regression model (1 degree of freedom) was used, including age, sex, body mass index (BMI), years of smoking, cigarettes consumed per day (cpd), and pack-years smoked (TI) as covariates; subsequently, Bonferroni correction was applied. The linkage disequilibrium structure and haplotype generation were performed with Haploview 4.2 software [10] according to the criteria of Gabriel SB et al. [17].

## 3. Results

### 3.1. Study Population

We found that 14.2% (*n* = 118) of the included population had a predominance of Caucasian (CEU) contribution (CTS = 51 and SWOC = 67). In contrast, in 85.8% (*n* = 712) of the population, the Native American (AME) component predominated (CTS = 248 and SWOC = 464). The 1 and 2 eigenvectors in the principal component analysis of all participants were considered (Figure 1).

The youngest group was SWOC with Caucasian predominance (56 years old), and fewer than 40% were men; however, this group had the highest amount of cigarette consumption (cpd = 40), and therefore, its packs-years smoked was higher (TI = 74) than that of the CTS group (*p* < 0.004).

Regarding the age of onset of tobacco smoking, the SWOC group with a Native American predominance began smoking at a later age (17 years) than the other three groups that started at 15 years old. As expected, the CTS groups had significantly lower lung function values than the SWOC groups, regardless of the ancestral contribution (Table 2).

According to the 2021 Global Strategy for Diagnosis, Management and Prevention of COPD (GOLD) [18], the classification of severity of airflow limitation in the group of COPD with high Caucasian component comprised 9.3% in GOLD I, 34.9% in GOLD II, 46.5% in GOLD III, and 9.3% in GOLD IV. The shares of those with a high Amerindian component in the COPD group were 9.9%, 24.9%, 49.7%, and 15.5% for GOLD I, II, III, and IV, respectively.

### 3.2. Alleles and Haplotype Analysis

Information about the minor alleles and their frequencies in the 452 SNPs in candidate genes analyzed in the population included in this study are included in Appendix A. We used age, sex, BMI, smoking time, cpd, and pack-years smoked as covariates for the allele association analysis. In a population with a high AME contribution, two SNPs (of 452 included in the microarray) in the *CYP2B6* and *CYP2A6* genes were associated with a lower risk of developing COPD; on the other hand, in the population with a CEU contribution, one SNP was associated with CYP2B6. The population with an AME contribution showed stronger significant *p*-values than individuals with CEU predominance (Table 3).

Haplotype analysis showed that SNPs with high linkage disequilibrium (LD) were present in the *CHRNA5-CHRNA3* loci. The largest block was in the group with CEU predominance, while in the group with a predominance of AME, the LD was lower and built two blocks, which were not associated (Figure 2); a block containing the alleles AGGAAGAGGGCA (rs8034191-rs2036527-rs684516-rs588765-rs17486278-rs5699207-rs637137-rs16969968-rs578776-rs105306-rs105306-rs105306-rs105306-rs104306-rs105306-rs105306-rs105306-rs105306-rs105306-rs104306-rs105306-rs105306-rs105306) exhibited a higher haplotype frequency (26%) for the CTS group than for the SWOC group (13%) with CEU predominance (>50%) in both groups; this was associated with a higher risk of COPD (*p* = 0.007, OR = 2.47, 95% CI: 1.3−4.8) (Table 4).

## 4. Discussion

This report assesses the influence of ancestral genetic contribution in an admixed Latin American population and genetic factors associated with COPD. In our study population, both SWOC groups had smoked longer than the respective CTS groups. The SWOC-AME group consumed a minor quantity of cpd; however, the onset of smoking was the most retarded compared to that of the other groups. According to “*Encuesta Nacional de Salud y Nutrición 2018-19*”, Mexican adults start smoking at 18.6 years [19], and our study population had a median starting age of 15 or 17 years old.

Previous reports describe that using cpd measures only one dimension of smoking exposure and intensity; in the current study, we calculated pack-years of smoking because it could assess the role of cumulative tobacco exposure [20]. However, our study population presented differences in the pack-years smoked; for this reason, we corrected the results by years of smoking, cpd, and pack-years smoked.

One report evaluated the ancestral genetic contribution and SNPs associated with the risk of COPD in a Latin American population characterized by a higher proportion of European ancestry (55.93% ± 6.89) and Mapuche ancestry (35.11% ± 8.54); in this study, the authors reported alleles in *PRDM15* associated with protection against COPD and SNPs in *PPP1R12B* associated with susceptibility [21].

In our population, with a high AME contribution, we found that SNPs in the *CYP2B6* and *CYP2A6* genes were associated with a lower risk of suffering COPD; of these two polymorphisms, the best studied is *CYP2A6*, which codes for the main hepatic enzyme metabolizing nicotine and cotinine [22]. Almost 90% of the nicotine that enters the body is transformed into cotinine through the participation of CYP2A6, while CYP2B6 metabolizes approximately 10% [23].

In the European Caucasian population, genetic variants were described in *CYP2A6* associated with cigarette consumption, and rs4105144 presented the highest association (*p* < 10^-12^) with cpd if the C allele was found. This polymorphism has a high linkage disequilibrium with rs1801272 (*CYP2A6 * 2*) and has been identified as an allele that reduces enzyme function [24]. We did not find haplotypes in *CYP2A6* or *CYP2B6* in our study population.

It should be noted that *CYP2A6* is a highly polymorphic gene. Studies in twins have revealed that alleles * 1B, * 1 × 2, * 2, * 4, * 9, and * 12 can explain up to 19% of the genetic variability in nicotine metabolism [25]; however, they are not the only variables of this gene. Therefore, it is crucial to evaluate them since smokers who have alleles associated with slow nicotine metabolism need fewer cigarettes to have the same reinforcing effects as those with alleles associated with rapid metabolism [26].

Notably, variants in *CYP2A6* contributed to the risk of developing COPD. This enzyme also metabolized several toxicant substances present in a cigarette puff, detoxifying some molecules and activating the toxic potential of other molecules that specifically induce lung injury, such as n-nitrosamines (4-(methylnitrosamino)-1-(3-pyridyl)-1-butanol (NNAL)) present in tobacco [27]. The presence of high levels of NNAL in urine was related to the exacerbation of COPD symptoms such as dyspnea, poor functional status, and more restricted activity [28]. Previously, in the Hainan population of China, polymorphisms in *CYP2B6* were related to susceptibility to COPD [29]. However, in our population, polymorphisms in *CYP2B6* and *CYP2A6* were associated with a lower risk of suffering from COPD; this indicates that in Latin America, Mexico had a minor prevalence of disease (7.8%) but had an annual cigarette consumption of 500–1499 per person [30], equal to Brazil, Uruguay, Chile, and Venezuela, which have a major prevalence at 10% [31].

Our findings reinforce the “Hispanic paradox” theory, which proposes “protection alleles” for COPD in some populations [32,33]; however, this theory has been questioned, primarily because of the complexity of the term “Hispanic”, even proposing that although it may have a genetic basis, the main reason could be the diet of these populations, which is characterized by abundance of legumes [32]. Several studies have reported that the consumption of antioxidants (flavonoids such as quercetin, vitamin E, and vitamin C) contained in food provides a protective effect against oxidants and free radicals produced by tobacco smoking [34,35,36]. Among other findings, legume intake is inversely associated with serum levels of adhesion molecules (E-selectin, sICAM-1, and sVCAM-1) and inflammatory biomarkers (C reactive protein, Tumor Necrosis Factor, Interleukin-6) [37]. In the Asian population, it has been found that a high consumption of soybeans is associated with a lower risk of COPD [38]. Our results provide evidence that COPD is a complex disease in which environmental factors (cigarette consumption, type of food, others) are involved; however, genetic factors located on chromosome 19 are essential and contribute to COPD.

Quitting smoking is crucial to prevent COPD progression; however, it is not always easy; COPD patients with moderate or high nicotine dependence (4 to 10 Fagerström Test for Nicotine Dependence, FTND) present more significant airway obstruction than those who have mild dependence (0 to 3 points in FTND) [39]. Some genetic variants (e.g., *AGPHD1, IREB2, HHIP*, and *FAM13*) have been associated with COPD regardless of cigarette consumption, while other polymorphisms in *CHRNA3* are associated with COPD due to cumulative exposure in smokers [40].

Our population haplotype analysis showed different blocks depending on the ancestral contribution; those with a Caucasian predominance had a higher risk (OR = 2.47) of suffering from COPD when a 93 kb haplotype was found in the *CHRNA5-CHRNA3* genes (chromosome 15); however, in the group with a high AME contribution showing linkage disequilibrium, it did not have an associated haplotype.

The subunits encoded by CHRNA3 and CHRNA5 can also be expressed in airway cells (bronchial epithelial cells, neutrophils, macrophages, monocytes, and lymphocytes); these receptors are activated when nicotine enters the lung, causing the release of proteases, proinflammatory cytokines, and oxidizing substances responsible for lung remodeling. After continuous exposure to cigarette smoking, there is a greater likelihood of lung damage [41].

The *CHRNA3* and *CHRNA5* genes participate in the mechanism of nicotine addiction and lung damage. The risk haplotype reported in our study includes rs16969968 and rs1051730. The first SNP has strong evidence in diverse populations associated with nicotine addiction, cpd, and COPD. A meta-analysis that included 34 studies (case-control, families, and cohorts) reported that the A allele (rs16969968) is associated (*p* = 5.96E-31) with a higher risk of cigarette consumption in the European population [42]. This polymorphism was also associated with airway obstruction, regardless of cigarette consumption, in seven cohorts analyzed in the European population [43].

On the other hand, rs1051730 was evaluated in smokers and nonsmokers in Denmark; in the group of smokers, the TT genotype was associated with a risk of COPD in stages of greater severity (GOLD III-IV stage) [44]. In a Dutch cohort of smoking individuals, T allele carriers (rs1051730) had an increased risk of bronchial obstruction and emphysema [45]. A copy of the T allele increases consumption by 1.2 cigarettes a day, suggesting that this region of chromosome 15 has an important function in smokers’ nicotine levels [46].

In a Russian population that included COPD patients and clinically healthy individuals (26% nonsmokers), the AAG haplotype formed by *CHRNA3/A5* (rs16969968-rs1051730-r6495309) was more frequent in COPD patients than in the control group [47]. In the Mexican Mestizo population, we identified, through a multistage genetic association study, polymorphisms in *CHRNA5* (rs16969968 and rs17408276) that are associated with cigarette consumption [48]. This report identifies a haplotype in the *CHRNA5* and *CHRNA3* genes in high linkage disequilibrium associated with a higher risk of COPD (OR = 2.47 95% CI 1.3–4.8).

Our study is not exempt from limitations. COPD is a complex polygenic disease and varies depending on environmental exposure, genetic variants associated with COPD risk, epigenetic mechanisms, etc. However, according to our results, ethnicity can influence the pathology. Although pack-years of smoking is a good tobacco smoking pattern indicator, the ideal would have been to have biochemical markers of nicotine such as serum cotinine or carbon monoxide exhaled and employed like a covariable.

Our results provide evidence of the importance of the ancestral component of COPD; several studies have reported the necessity of considering ancestral contributions to COPD risk to prevent the disease and identify individuals at a markedly increased risk [49,50]. We showed that existing alleles in *CYP2B6* and *CYP2A6* were associated with a minor risk of COPD; in the Mexican population with a high contribution of CEU and the presence of the haplotype in *CHRNA5*-*CHRNA3*, there was a predisposition to an increased risk of disease (Figure 3). These genetic regions are of importance in COPD and differ according to population ancestry.

## 5. Conclusions

In Mexican-Mestizo smokers, there are SNPs associated with a lower risk of COPD in genes that encode proteins responsible for nicotine metabolism. Nevertheless, individuals with a high Caucasian component harboring a haplotype in the *CHRNA5-CHRNA3 loci* have a COPD risk that is twice as high.

## Figures and Tables

**Figure 1 diagnostics-11-00599-f001:**
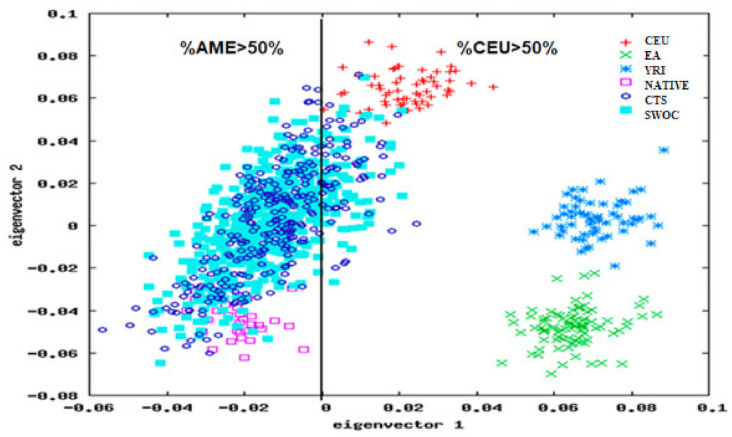
Principal component analysis in CTS and SWOC groups. Reference populations: Caucasian (CEU), East Asian (EA), and African (YRI) populations from the international HapMap project and Amerindian population of Mexican Genome Diversity Project (NATIVE). AME: high Amerindian component. CEU: high Caucasian component.

**Figure 2 diagnostics-11-00599-f002:**
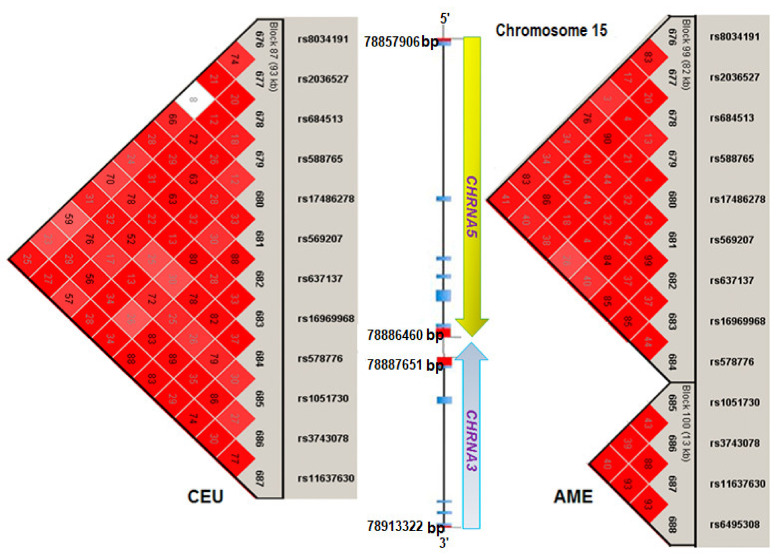
Haplotype in the *CHRNA5-CHRNA3* loci between CTS and SWOC with a high Caucasian component (CEU) or a high Amerindian contribution (AME). Bp: base pairs. The analysis was performed with Haploview 4.2 [11]. CEU: high Caucasian component. AME: high Amerindian component.

**Figure 3 diagnostics-11-00599-f003:**
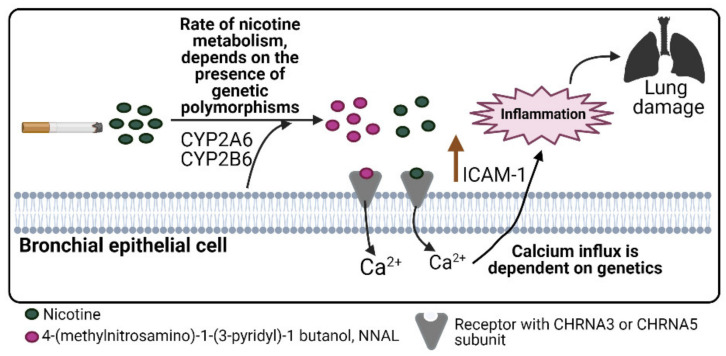
Participation of CYP2A6, CYP2B6, CHRNA3 and CHRNA5 in COPD. Created with BioRender.com.

**Table 1 diagnostics-11-00599-t001:** Genes included in this study and the number of single nucleotide polymorphisms (SNPs) in each chromosome.

Chr	Genes	Number of SNPs
1	*EPHX1* and *IL6R*	24
2	*GNLY*, *SERPINE2*, and *SFTPB*	62
3	*PDZRN3*	30
4	*GYPA* and *HHIP*	30
5	*ADAM19* and *ADRB2*	51
6	*GPR126* and *TNF*	32
7	*HIP1*, *IL6*, and *SERPINE1*	37
8	*CHRNB3*	4
9	*TNFSF8*	24
10	*SFTPD*	15
13	*FOXO1*	13
15	*CHRNA3*, *CHRNA5* and *CHRNB4*	16
16	*MMP15*	15
19	*CYP2A6*, *CYP2B6*, *EGLN2*, *GPATCH1*, and *TGFB1*	61
20	*ADAM33*	12
21	*ADARB1*	26

Chr; chromosome.

**Table 2 diagnostics-11-00599-t002:** Demographic and clinical variables of the study population.

Variable	CEU (*n* = 118)		AME (*n* = 712)	
	CTS-CEU (*n* = 51)	SWOC-CEU (*n* = 67)	CTS-AME (*n* = 248)	SWOC-AME (*n* = 464)
Age (years) ^1^	63 (50–80)	56 (52–71)	65 (35–84)	65 (48–76)
Sex, male (%) ^1,2^	43.7	39.4	42.0	50.0
BMI (kg/m^2^) ^2^	25 (17–35)	30 (22–33)	25 (18–36)	30 (21–36)
Time smoking (years) ^1,2^	40 (10–68)	44 (37–56)	41 (10–62)	45 (10–67)
Cpd ^2^	20 (10–80)	40 (20–60)	20 (10–80)	17 (10–60)
Onset of smoking (years) ^2^	15 (8–30)	15 (12–16)	15 (7–60)	17 (11–50)
Pack-years smoked ^1,2^	36 (5–170)	74 (44–168)	40 (6–200)	33 (6–132)
FVC (%) *^,1,2^	82 (18–146)	90 (66–92)	86 (35–162)	65 (20–114)
FEV1 (%) *^, 1,2^	53 (21–110)	80 (65–81)	58 (15–119)	65 (62–120)
FEV1/FVC *^, 1,2^	53 (25–69)	71 (70–75)	54 (19–69)	73 (70–84)
Population ancestry (%)				
Caucasian ^2^	64.2	57.0	36.3	37.0
Amerindian ^1^	31.5	37.8	61.0	61.4
African	1.0	1.1	0.4	0.4
East Asian	1.5	1.8	0.6	0.6

Showing median (minimum−maximum value) or percentage. BMI: body mass index. Cpd: cigarettes consumed per day. FVC: forced vital capacity. FEV1: forced expiratory volume in the first second (* postbronchodilator spirometry values). CTS-CEU: COPD patients with a high Caucasian component. SWOC-CEU: smokers without COPD and high Caucasian component. CTS-AME: patients with COPD and a high Amerindian component. SWOC-CEU: smokers without COPD and high Amerindian component. ^1^, variable with *p* < 0.05 in CTS-CEU vs. SOWC-CEU. ^2^, variable with *p* < 0.05 in CTS-AME vs. SOWC-AME.

**Table 3 diagnostics-11-00599-t003:** Results of the allele association in CTS vs. SWOC according to the ancestral component.

SNP	Gen	Risk Allele	CEU		AME	
			*p*	OR	*p*	OR
rs10426235	*CYP2B6*	A	6.83E-03	0.221	9.17E-10	0.044
rs4105144	*CYP2A6*	A	6.43E-01	0.774	2.11E-03	0.485

CEU: high Caucasian component. AME: high Amerindian component.

**Table 4 diagnostics-11-00599-t004:** Haplotype analysis in the groups with a high Caucasian component.

Haplotype * *(CHRNA5-CHRNA3)*	Haplotype Frequency (%)		*p*
	CTS (*n* = 51)	SWOC (*n* = 67)	
AGGAAGAGGGCA	26	13	0.007
AGCGAATGAGGG	30	39	0.174
GAGGCGAAGACA	23	20	0.667

* Haplotypes with a frequency >10% in the study population.

## Data Availability

Data are available and may be obtained by contacting the corresponding author.

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
