# Peer review of "Genetic Factors Associated with COPD Depend on the Ancestral Caucasian/Amerindian Component in the Mexican Population"

_diagnostics, 2021, doi:10.3390/diagnostics11040599_

Round 1

Reviewer 1 Report

Dear Editor,

The study design is very interesting and well designed and done but the results are obviously expected, nothing new and since my experience bad explained, ther is a very poor a no explained disccussion.

  1. I think there is a big lack of references in healthy smokers population, without other interactions.
  2. The results are due for the main impact of CYP2A6 and CYP2B6 in nicotine dependence (instead of corrections) but in the discussion they only talk about other studies in a descriptive way. Please, you must interprete your results.
  3. It is neccesary complete the p values in Table 1 to have clear the basic associations.
  4. It is not clear if their results are due to dependence and PYS or has an explanation of a biological point of view. Add this information.
  5. It is true that without CO levels you can not be sure that your population is well phenotyped.

Author Response

  1. I think there is a big lack of references in healthy smokers population, without other interactions.

Thank you for comment; the smokers without COPD had evaluated by specialized pulmonologists and had with a spirometry data. In study population we used years of smoking, cigarettes consumed per day (cpd) and pack-years smoked like covariables for corrected our results for these variables.

2. The results are due for the main impact of CYP2A6 and CYP2B6 in nicotine dependence (instead of corrections) but in the discussion they only talk about other studies in a descriptive way. Please, you must interprete your results.

You are right, CYP2A6 and CYP2B6 participated in nicotine dependence, however, these enzymes are expressed in lung; in lines 227-238 we add evidence of role of CYP2A6 and CYP2B6 in the COPD; we integrated our results in figure 3.

3. It is neccesary complete the p values in Table 1 to have clear the basic associations.

Thank you; table 1 show the SNPs included per chromosome; in this table is not possible calculated p-value.

4. It is not clear if their results are due to dependence and PYS or has an explanation of a biological point of view. Add this information.

In lines 227 to 238 we add evidence of contribution of CYP2A6 and CYP2B6 in COPD at lung level.

5. It is true that without CO levels you can not be sure that your population is well phenotyped.

Thank you, the cpd was used like a simple measure that captures in part the genetic association (PMID: 22102629); however, an easily measurable biomarker would have guaranteed our finding without the need to use covariates such as years of smoking, cigarettes consumed per day (cpd) and pack-years smoked

Reviewer 2 Report

I confirmed that the points that were pointed out by reviewer has been corrected in this manuscript . This work is interesting to respiratory disease field.

Author Response

Thank you for the comments

Reviewer 3 Report

In the present study, the authors investigated the ancestral contribution to COPD incidence. They were using a customized genotyping array of SNPs.  The aim of the study was to identify  SNPs associated with COPD secondary to tobacco smoking in the Mexican population of different origins. 

Results and Discussion are very extremely poorly written, without any significant information. Although this is the re-submission, I would not recommend the acceptance of the manuscript.  In general language and style should be significantly improved - punctuation marks are used in a totally wrong way!

In general, it is not news that complex diseases, like COPD, are influenced by ancestral components. For me, it does not make any sense to mix genetic background and motivation for smoking cessation!? 

Also, I would say that there are too few results for the journal with IF 3.1 (Q1).

Author Response

Thank you for the comments, we corrected results and the discussion, add figure 3 with our results in the context of COPD (lines 303-307). Now the paper has style correction.  We contextually our data and compare with other populations. In figure 3 we explain the results obtained and the possible participation in COPD.

Round 2

Reviewer 1 Report

NO COMMENTS

This manuscript is a resubmission of an earlier submission. The following is a list of the peer review reports and author responses from that submission.

Round 1

Reviewer 1 Report

Pérez-Rubio G et al. “Genetic Factors Associated with COPD Depend on the Ancestral Caucasian/Amerindian Component in the Mexican Population”

The authors focus on racial difference of COPD morbidity rate and explored genetic factors related to the risk of COPD by using SNPs. As a result, the authors showed nicotine associated genetic changes could contribute to the risk of COPD. The study was well performed and the findings are interesting, however, there are several major concerns with this work as outlined below.

(Major comments)

  1. How did authors determine the selected candidate genes (p3, line93-96)? The detailed reason should be stated.
  2. I can not find the result of the selected candidate genes other than CYP2, CHRNA. The author should describe and discuss the results.
  3. Recently, it is reported that COPD is not simply a disease that is caused by smoking and early life influences on the development of COPD. Therefore, the result may be restricted to traditional mechanism of COPD. The author should add this point as limitation or discussion.
  4. How do the result be utilized in clinical setting? The author had better to show the outlook.

Reviewer 2 Report

Nothing specific